# OpenReview forum: "CTFusion : A CTF-based Benchmark for LLM Agent Evaluation"
_ICLR.cc/2026/Conference — Submitted to ICLR 2026_

### Official Review · Reviewer_xRZ7 · 2025-10-30

**Soundness:** 3
**Presentation:** 3
**Contribution:** 3
**Rating:** 4
**Confidence:** 4

**Summary:**

Existing Capture The Flag (CTF) benchmarks for evaluating Large Language Model (LLM) agents in cybersecurity are unreliable due to widespread data contamination and potential cheating from publicly available solutions. To address this, the paper introduces CTFUSION, a novel streaming evaluation framework built on LIVE CTFS competitions, which uses virtualization and aggregation to ensure isolated agent interaction and minimize impact on real contests. Experiments show that static benchmarks significantly inflate agent performance (e.g., 14.4% success rate) compared to CTFUSION's live evaluation (6.3%), validating CTFUSION as a robust solution for assessing genuine LLM agent capabilities in cybersecurity.

**Strengths:**

1. The design is well-articulated: it employs a two-tier gateway with an MCP server and proxy that enable per-agent view isolation (each agent sees all challenges as unsolved until it solves them), and aggregation (ensuring each challenge is credited only once to the contest).
2. The authors implement the full system and plan to open-source it.
3. The paper tackles an important issue at the intersection of LLM agents and cybersecurity.
4. A particularly compelling part of the work is the analysis of the D-CIPHER-WEB agent. By enabling a simple web-search step, they show that the agent’s success rate on the static bench nearly doubles, and they back this up with logs: “71 cheating attempts” were logged, including 63 direct flag-copy events.

**Weaknesses:**

1.  Several important claims are not fully substantiated. For example, the paper states that CTFUSION has “negligible impact on real competitions”, but provides no empirical data to support this. There is no measurement of server load, latency, or effect on other teams. The assertion of “minimal configuration changes” in applying CTFUSION is plausible but not demonstrated beyond five contests. Likewise, the claim that static benchmarks are unreliable rests on the observed performance gap, but the paper does not isolate the causes. The authors themselves note two factors—task difficulty and contamination—but do not quantify their contributions. It remains possible that the live CTFs tested were simply harder or less representative than the static tasks. Without a controlled ablation (e.g. using an LLM with an earlier knowledge cutoff, or filtering static tasks with known write-ups), it is hard to attribute the gap primarily to data leakage.
2. The paper focuses on aggregate success rates, but offers little deeper insight. For instance, there is no breakdown of results by challenge category (crypto vs pwn vs web, etc.), nor analysis of which particular tasks were solved or failed.
3. The use of a fixed pass@3 metric and $3 cost cap is reasonable, but details are sparse. For example, how random are the results? Did the authors run multiple trials to compute confidence intervals? None are reported, so it's unclear how much variance there is.
4. Novelty is limited.  The system design reuses well-known components (MCP, Docker, API proxies). The paper does not introduce new learning techniques or benchmarks, but rather repurposes existing CTF infrastructure in an incremental way.

**Questions:**

1. Can you provide evidence that tasks in the NYU CTF BENCH actually appeared in the LLMs’ training data or public write-ups? For example, did you check which static challenges have publicly available solutions and correlate that with agent success? This would clarify whether data leakage is the main cause of the performance gap (rather than inherent difficulty).
2. You use pass@3 with a $3 cost cap for each attempt (Sec.5.1). How sensitive are your results to these choices? Did you try different k (e.g. pass@1 vs pass@5) or cost limits to see if the relative performance changes? Also, do agents terminate upon a correct flag or always use all k attempts? Some clarity on agent stopping criteria would be helpful.

---

> ### Author Response · Authors · 2025-11-19
> **Response to Weaknesses**
>
> We thank the reviewer for carefully highlighting these weaknesses and for the constructive suggestions on how to strengthen the paper, especially regarding empirical support for our claims, deeper analysis, and clarification of evaluation choices. We address each point in turn below.
>
> ### Response to Weaknesses1
> 1) Impact on real competitions.
>
> CTFusion interacts with each challenge in a very constrained way: for a given CTF, all six model-agent pairs share a single team account, and each challenge is fetched once, followed by number of flag submissions. In practice, this results in strictly fewer HTTP requests than a typical human team that iteratively refreshes challenge pages, downloads artifacts multiple times, and submits flags repeatedly.
>
> 2) Isolating contamination vs difficulty.
>
> To better separate the effect of contamination from inherent task difficulty on NYU-CTF-bench, we have added an ablation using D-CIPHER with contamination controls. Concretely, we (i) ran D-CIPHER under a stricter knowledge-cutoff / retrieval configuration, and (ii) filtered static tasks that have known public write-ups or direct package-level access. Additional experiments along this line are currently in progress, and we will provide further comments as soon as the results are finalized.
>
> ### Response to Weaknesses2
> We agree that aggregate success rates alone provide limited insight. In the revised paper, we add a dedicated subsection on F**ailure Causes and Categories Analysis**, which includes:
>
> - A breakdown of **success and failure by challenge category**.
> - Analysis of **why agents fail** on specific category.
> - A summary of the **most common fail reason** across agents.
> - Identification of **problem types where agents perform relatively well** versus those where they systematically struggle.
>
> ### Response to Weaknesses3
> We appreciate the request for more detail on randomness and variance. Our current evaluation uses a fixed pass@3 metric with a $3 cost cap per challenge run, and each agent–model combination is executed once per challenge with a single sampling configuration.
>
> We did not run enough repeated trials to compute robust confidence intervals for all settings, mainly due to time constraints on large-scale CTF runs. We have clarified this limitation in the revised paper and explicitly state that our current results should be interpreted as point estimates rather than fully characterized distributions(5.1 Evaluation Targets and Metrics).
>
> ### Response to Weaknesses4
> The novelty of our work does not lie in introducing a new learning algorithm or building an entirely new CTF platform, but rather in how these pieces are combined to enable a new evaluation paradigm:
>
> 1. Problem setting shift (static → live streaming evaluation).
>
> We empirically demonstrate limitations of static CTF benchmarks (contamination, leakage, and cheating) and then formalize ongoing live CTFs as a streaming benchmark via CTFusion. To our knowledge, this is the first framework that systematically turns live competitions into a reproducible evaluation pipeline for LLM cybersecurity agents.
>
> 2. Two-layer gateway design (read via MCP, write via proxy) with isolation and aggregation.
>
> CTFusion enforces isolated views per agent (through virtualization and MCP-based tools) while routing all flag submissions through a single proxy account that aggregates scores. This layered design (isolation + aggregation) is crucial: it preserves fairness and reproducibility while ensuring that we do not distort the real scoreboard or leak challenge information between agents.
>
> 3. Evaluation protocol and reproducibility contribution.
>
> The framework specifies a concrete protocol for running agents in live events (budgeting, stopping criteria, isolation, logging) and provides open-sourced code, prompts, and reconstruction scripts so that future work can reproduce our setting or plug in new agents/models without re-engineering the infrastructure.
>
> In short, our contribution is conceptual and methodological: we introduce and validate a live, leakage-resilient evaluation protocol for LLM agents in CTFs, built from well-known components but engineered into a coherent, practically deployable framework.

---

> ### Author Response · Authors · 2025-11-19
> **Response to questions**
>
> ### Response to Question 1
> We agree that concrete evidence of data leakage is important.
>
> Direct evidence from agent logs.
>
> - When we extended D-CIPHER with web access (D-CIPHER-WEB), we observed explicit cheating behavior in the evaluation logs
> - 71 cheating attempts in total, consisting of 63 direct flag-copy submissions and 8 cases where publicly available write-ups were copied or closely followed.
> - Table 1 and Appendix A includes representative examples, where agents retrieve flags or exploit scripts directly from public sources and submit them without performing the actual exploit steps.
>
> Open-source accessibility and direct dataset access.
>
> - NYU-CTF-bench is fully open-sourced (code and challenge artifacts), and all of its challenges have publicly available solutions. In several cases, the agent directly interacted with these artifacts.
> - D-CIPHER-WEB installed the nyuctf Python package during evaluation and used the CTFDataset / CTFChallenge APIs to enumerate challenges and read flags directly (e.g., the “1nsayne” task in Figure 3).
> - In other tasks, the agent fetched raw files or solutions from GitHub repositories and public write-ups, then immediately submitted the derived flags (as summarized in Table 1).
> - These behaviors show that, for NYU-CTF-bench, solve code, flags, and datasets are reachable through public channels that LLM agents can access at evaluation time, structurally increasing the chance of contamination and opportunistic cheating.
>
> Temporal overlap with training data (indirect evidence).
>
> - NYU-CTF-bench consists of challenges from roughly 2017–2023, many of which have long-standing public write-ups. Commercial LLMs we evaluate have knowledge cutoffs in 2024–2025, making it highly plausible that some of these challenges and solutions are present in their pre-training corpora. While we cannot inspect proprietary training sets directly, we discuss this temporal overlap in the revision as indirect evidence that contamination is likely.
>
>
> ### Response to Question 2
> 1. **Sensitivity to k (pass@1 vs pass@3)**
>
> We evaluated pass@1, pass@2, and pass@3 across all model–agent combinations. As expected, success rates increase monotonically with larger k. The following table summarizes the results:
>
> | Model + Solver     | pass@1 | pass@2 | pass@3 |
> |--------------------|--------|--------|--------|
> | gpt + enigma       | 15.56% | 17.78% | 19.44% |
> | gpt + dcipher      | 10.56% | 12.22% | 14.44% |
> | claude + enigma    | 7.22%  | 11.67% | 12.22% |
> | claude + dcipher   | 8.89%  | 10.00% | 10.56% |
> | gemini + enigma    | 11.11% | 15.00% | 17.22% |
> | gemini + dcipher   | 7.22%  | 11.11% | 12.78% |
> |
>
> Although the absolute values change, the relative ordering of the top-performing systems is stable: the same three combinations consistently occupy the 1st, 2nd, and 3rd positions for all k.
>
> 2. **Sensitivity to the $3 cost cap.**
>
> We did not explore cost limits higher than USD 3. In our qualitative inspection of trajectories, cases that would have benefited from significantly more than USD 3 of API calls were extremely rare. In practice, almost all successful runs terminated well below this threshold. Thus, the USD 3 limit acts more as a conservative upper bound than an active constraint in most episodes.
>
> There are three stopping conditions in our system:
>
> - Solved: a correct flag is submitted.
> - Cost threshold: the USD 3 limit is reached.
> - Self-termination: the agent decides to stop trying.
>
> To help readers understand, we have added clarity to the termination conditions in the revised version.(5.1 Evaluation Targets and Metrics)

---

> ### Author Response · Authors · 2025-12-03
> **Response to controlled ablation**
>
> We have added Section 3.3 “Controlled Ablation on NYU CTF Bench” and Table 2, where we evaluate a D-CIPHER-NO-CHEAT variant that differs from D-CIPHER only in a single prompt instruction: “Do not use any pre-trained knowledge or direct information about this assignment. Rely only on your own reasoning.” This is a prompt-level controlled ablation in the spirit of the reviewer’s suggestion (it explicitly asks the model not to reuse benchmark-specific prior knowledge), while keeping the underlying model, agent, and evaluation pipeline fixed.
>
> On the static NYU CTF BENCH, this controlled ablation reduces pass@3 performance as follows:
>
> - GPT-4.1: 14.44% → 9.44%
> - GEMINI 2.5-FLASH: 12.78% → 10.00%
>
> Across these two models, D-CIPHER’s average success rate drops from 13.6% (49/360) to 9.7% (35/360), a 3.9% reduction.
> Thus Contamination-related behavior clearly has a non-negligible effect: removing the ability to freely rely on pre-trained benchmark knowledge reduces static performance by about one third.

---

### Official Review · Reviewer_NbCu · 2025-11-01

**Soundness:** 3
**Presentation:** 3
**Contribution:** 3
**Rating:** 6
**Confidence:** 3

**Summary:**

This paper highlights the data contamination and evaluation bias risks in existing CTF-based benchmarks for assessing LLM cybersecurity agents. To verify these concerns, the authors conduct controlled experiments comparing model performance on static benchmarks versus unseen challenges, confirming significant contamination effects. To address this, the paper introduces CTFusion, a new framework that evaluates models through live CTF competitions, ensuring fairness and preventing leakage from pre-existing datasets or write-ups. Experiments on five real-world live CTFs show that model performance drops by nearly half compared to static benchmarks, revealing how current evaluations substantially overestimate model capabilities. The proposed CTFusion framework establishes a reproducible, isolation-based, and dynamic evaluation pipeline, setting a more reliable foundation for future LLM agent research in cybersecurity.

**Strengths:**

1. Strong motivation: Identifies a previously overlooked issue—benchmark contamination and fairness in LLM CTF evaluation;
2.  Robust engineering design: Virtualization, proxy aggregation, and live integration are elegantly executed;
3. Strong empirical findings: Demonstrates clear performance overestimation in static benchmarks (e.g., GPT-4.1 drops from 19.4% → 9.7%);
4. Practical impact: Encourages the community to move towards live and transparent benchmarking;
5. Reproducibility: Open-sourced framework with standardized interfaces to CTFd and MCP APIs;

**Weaknesses:**

1. One major consequence of data contamination is the loss of fairness in evaluation. Could the authors report the correlation between model rankings on your benchmark (CTFusion) and those on prior CTF benchmarks? Such an analysis would clarify whether the contamination issue also affects relative ranking stability across benchmarks;
2. The evaluation exclusively relies on three closed-source commercial models without including open-source alternatives such as Llama, Qwen, or DeepSeek. This omission limits the generalizability of findings and undermines the data contamination hypothesis, as open-source models with transparent training data could serve as crucial controls;
3. The paper lacks detailed error analysis and failure case studies that would provide actionable insights for future research. While the authors report aggregate success rates across models and agents, they do not systematically analyze why agents fail on specific challenges, what types of errors are most common, or which vulnerability categories pose the greatest difficulties;

**Questions:**

See the Weaknesses part

---

> ### Author Response · Authors · 2025-11-19
>
> We thank the reviewer for carefully raising concerns about potential limitations of our work and for the constructive suggestions on how to strengthen the paper, particularly regarding fairness, model coverage, and error analysis. We address each of these points below.
>
> ### Response to Weaknesses1
> As shown in Figure 6 of the paper, we examined the consistency of model rankings between the prior static CTF benchmark (NYU-CTF-bench) and our live CTFusion evaluation. On both NYU-CTF-bench and LiveCTF with CTFusion, the ranking is 1)GPT, 2) Gemini, 3) Claude. Since the relative ordering of the three models is identical across both benchmarks, the relative ranking of the models remains stable in our current setting.
>
> ### Response to Weaknesses2
> We agree that including open-source models such as Llama, Qwen, or DeepSeek would further strengthen the generality of our findings and could, in principle, help reason about contamination if their training data were fully transparent.
>
> However, to the best of our knowledge, these models do **not** provide a complete, enumerated list of all training datasets or a fully transparent corpus specification; they primarily state that they are trained on publicly available. Thus, in practice, the transparency of their training data with respect to potential CTF contamination is not fundamentally different from that of closed-source commercial models like GPT, Gemini, and Claude.
>
> ### Response to Weaknesses3
> We appreciate this suggestion and have expanded the evaluation section to include a more systematic error and failure analysis. In the revised paper, we add a new appendix titled “Failure Causes and Categories Analysis”, which provides:
>
> - A breakdown of **success and failure by challenge category**.
> - Analysis of **why agents fail** on specific category.
> - A summary of the **most common fail reason** across agents.
> - Identification of **problem types where agents perform relatively well** versus those where they systematically struggle.

---

### Official Review · Reviewer_bsbd · 2025-11-04

**Soundness:** 3
**Presentation:** 2
**Contribution:** 2
**Rating:** 2
**Confidence:** 4

**Summary:**

The paper presents

1. A study on potential contamination/cheating in commonly used CTF benchmarks
2. A new framework that allows to make easy to run on currently ongoing challenges on CTFd, the biggest site of CTF competitons

For 1, the authors modify D-ciper a CTF agent by forcing it to start every agent run with a google search. They then analyze how often the agent "cheats", i.e., retrieves the solution instead of solving the challenge themselves.

For 2, the authors create a new framework that allows to conveniently evaluate multiple agents on ongoing CTFd challenges without being disruptive to the CTFd leaderboards.

**Strengths:**

Tapping into live challenges to evaluate LMs on CTF skills without any possible contamination or cheat issues is a relevant contribution. Based on the details in the paper, the implementation of this novel evaluation framework seems to be well-engineered and thought-through in particular giving thought to avoid disrupting the CTFd leaderboards disproportionately.

The studies on cheating are an interesting and novel way to essentially determine upper limits of possible cheating.

The paper is overall well written, though some section might benefit from focusing on the big picture over implementation details.

**Weaknesses:**

**Studies on models cheating**

The study essentially seems to be giving an "upper limit" to cheating , as agents are specifically told to "google for anything that might be related" and generally almost incentivizing the model to "cheat". I think this is an interesting idea, but it doesn't necessarily quantify how much cheating occurs in current benchmark evaluations. For example, benchmark evaluators might specifically ask models not to cheat by asking them to not read any writeups and only search for technical details (or not to use a search engine at all), thereby at the very least significantly reducing this form of cheating.
I assume that D-cipher technically has the ability to search for things by using `wget` or similar, but just from the score difference with D-cipher-web, it definitely doesn't seem to look up solutions often.

I also wonder: If you can separate a cheating and non-cheating component on the D-cipher web run, couldn't you do the same for the default D-cipher and show how much of its performance is due to cheating in Fig 2.?

**Using CTFd for benchmarking**

The main weakness is taking this new datasource and obtaining interesting results about model behavior, ranking, etc. In other words: What can I learn from this new benchmark (or based on this new datasource)?
Clearly, existing benchmarks are somewhat contaminated by writeups and so it is not too surprising that scores differ (but scores are also vastly different between different CTF benchmarks anyway). Moreover, as the paper remarks itself, there are also other conflating factors other than just contamination/cheating. One possible direction the paper could explore is whether the model ranking based on LiveCTF is different than on other benchmarks, for example. One of the possible issues with the LiveCTF might be that there are only relatively few task instances available at any point, so thought would have to be given to statistical treatment.
If on the other hand, old task instances from past challenges are kept on being used, then the benchmark cannot add newly released models while staying completely contamination free, facing the same issues as other benchmarks.
Focusing the paper on these questions and on concrete takeaway messages would significantly strengthen the paper.

**Readability**

* Fig 4 has numbers for the arrows, but they don't seem to be references by the text.
* I found Fig 5 hard to read. At normal magnification, the legend for the color coding is very small and the hatching color patches barely visible. It would also be good to maybe show `n=` (number of instances in each of these datasets) in the title of each subplot. That would help to get a feeling of what uncertainties might be. One idea would be to use horizontal bar charts that way you can have the model/agent names spelled out on the y axis, much easier to read
* Fig 6 left I found pretty confusing at first, the only thing I immediately understood is that live ctfs are harder than nyu ctf. Wouldn't it be much easier to present this in another bar chart?
* Line 234: It took me quite some to understand statements like "two-tier gateway architecture (enforces) isolated view for each agent" means (this is mentioned again also in 273). Reducing jargon and implementation details and focusing on the bigger picture might make 4.1 + 4.2 an easier read, for example by starting with the reason why you only want to have one CTFd account and then explaining the basic principles of the implementation in simple language.

**Questions:**

* D-Cipher vs D-cypher web. I assume D-cipher technically already had that capability by using wget etc.? So the main difference with D-cipher web is  that you forced it to start with a search?
* Regarding the evaluation setup/implementation: Do certain challenges require a more complicated setup? E.g., having a specific docker container etc.? How is that handled automatically in this setup (since you download the problem using mcp it sounds like the docker container is always the same?)
* Line 416: How is difficulty determined? I.e., how do you know that the performance gap is partially due to difference in difficulty (or is that a hypothesis?)

---

> ### Author Response · Authors · 2025-11-19
> **Response to Weaknesses**
>
> We thank the reviewer for carefully raising these concerns about cheating behavior, the value of LiveCTF as a benchmark, and the readability of the paper. Your comments helped us clarify the scope of our claims, refine the experimental design, and improve the presentation. We respond to each point below.
>
> ### Response to Weaknesses1
>
> We agree that our D-CIPHER-WEB setup is best interpreted as providing an upper bound on how much cheating is possible under a permissive policy, rather than a direct estimate of how much cheating occurs in typical benchmark evaluations.
>
> - Relationship to the default D-CIPHER.
>     - As the reviewer notes, the default D-CIPHER technically could fetch external resources via tools such as wget, but in our experiments it rarely does so to retrieve complete solutions. This is reflected in the smaller performance gap between D-CIPHER and its web-augmented variant. We emphasize that the point of D-CIPHER-WEB is to expose structural vulnerabilities of static, open benchmarks (where flags and write-ups are one search away), rather than to say that all agents always cheat in practice.
>
> - Separating cheating vs. non-cheating components.
>     - For D-CIPHER-WEB on NYU-CTF-bench, we can indeed distinguish “cheating” from “non-cheating” episodes based on logs: runs where the agent directly copies flags or closely follows a write-up/solve script are tagged as cheating.
>
>     - To better separate the effect of contamination from inherent task difficulty on NYU-CTF-bench, we have added an ablation using D-CIPHER with contamination controls. Concretely, we (i) ran D-CIPHER under a stricter knowledge-cutoff / retrieval configuration, and (ii) filtered static tasks that have known public write-ups or direct package-level access. Additional experiments along this line are currently in progress, and we will provide further comments as soon as the results are finalized.
>
> ### Response to Weaknesses2
> We appreciate the question "what can I actually learn from this new data source?" and agree that the paper should sharpen its takeaways.
>
> Beyond confirming that static benchmarks overestimate absolute performance, the revised paper emphasizes three concrete findings:
> 1. Absolute vs. relative performance.
> On NYU-CTF-bench, the model ranking is GPT > Gemini > Claude; on LiveCTF with CTFusion, we observe the same ranking. This shows that while static benchmarks substantially inflate absolute success rates, the relative ordering of the three models remains stable in our current setting. We now highlight this explicitly as a core takeaway.
>
> 2. Challenge-type–specific weaknesses.
> We added a deeper analysis (as also mentioned to other reviewers) that breaks down performance by category (pwn, web, crypto, reversing, etc.) and discusses which challenge types are systematically hard for agents in live settings.
>
> 3. Contamination vs. reusing past challenges.
> CTFusion is a framework for turning ongoing CTFs into evaluation benchmark, not a fixed static dataset. In this work, we only evaluate on challenges during live.
>
> ### Response to Weaknesses3
> Thank you for the detailed, actionable suggestions on readability. We have revised the figures and text accordingly

---

> ### Author Response · Authors · 2025-11-19
> **Response to questions**
>
> ### Response to Question 1
> yes, your understanding is essentially correct.
>
> The default D-CIPHER already has the technical ability to fetch external resources (e.g., via wget) if the agent decides to do so.
>
> D-CIPHER-WEB adds: An explicit web search tool, and A prompt that forces the agent to begin with a search and then follow relevant search results when appropriate.
>
> In other words, the main difference lies in the prompting and tool interface: D-CIPHER-WEB makes web search the default first step and structurally encourages retrieval from public sources, whereas the default D-CIPHER relies on the agent to decide when to use such capabilities.
>
> ### Response to Question 2
> Our evaluation intentionally does not attempt to replicate challenge-specific runtime environments such as custom remote Docker images.
>
> We run each agent (ENIGMA and D-CIPHER) inside its official, publicly released Docker container, using one isolated container per agent–model pair for every challenge attempt.Even if a problem requires a more complex setup, the environment is initialized exactly the same as in the beginning, and all subsequent operations are handled according to the LLM’s decisions.
>
> ### Response to Question 3
> We use the standard CTFtime weight metric—which is computed from global participation and placement statistics and is widely adopted in the CTF community—as our primary quantitative proxy for event difficulty. The weights of the NYU-CTF-Bench events range from 21.8 to 59 (mean 31.64), whereas our LIVE CTF events range from 24.48 to 69.35 (mean 39.64), so the two distributions are broadly comparable. This suggests that the two benchmarks lie in a similar overall difficulty regime, although LIVE CTF is, on average, somewhat more difficult according to this metric. But roughly 2× performance gap remains clearly large even after taking difficulty into account. We have also added these details into both the main text and the appendix.

---

> ### Author Response · Authors · 2025-12-03
> **Response to Studies on models cheating**
>
> We appreciate the reviewer’s suggestion to more directly quantify how much of D-CIPHER’s performance on NYU CTF BENCH can be attributed to cheating / contamination-like behavior, and to compare against a setting where the agent is explicitly discouraged from cheating.
>
> In the revised manuscript we have added Section 3.3 “Controlled Ablation on NYU CTF Bench” and Table 2, where we introduce a variant D-CIPHER-NO-CHEAT. This agent is identical to D-CIPHER in tools, interaction protocol, and cost budget; the only change is an additional instruction appended to the planner’s system prompt for every challenge: “Do not use any pre-trained knowledge or direct information about this assignment. Rely only on your own reasoning.”
>
> On NYU CTF BENCH (180 problems), we obtain the following pass@3 results:
> - GPT-4.1 + D-CIPHER: 14.44% (26/180)
> - GPT-4.1 + D-CIPHER-NO-CHEAT: 9.44% (17/180)
> - GEMINI 2.5-FLASH + D-CIPHER: 12.78% (23/180)
> - GEMINI 2.5-FLASH + D-CIPHER-NO-CHEAT: 10.00% (18/180)
>
> Averaged across these two models, the original D-CIPHER solves 13.6% of problems (49/360), whereas the no-cheat variant solves 9.7% (35/360), a 3.9% drop, or about 29% relative reduction. Because all other aspects of the setup are held fixed, this ablation provides a controlled estimate of how much the agent’s performance depends on being allowed to reuse benchmark-specific prior knowledge. In particular, it suggests that roughly one third of D-CIPHER’s success on NYU CTF BENCH is plausibly driven by contamination-related behavior, such as recalling or reconstructing known solutions, rather than de-novo problem solving.

---

### Official Review · Reviewer_N8G9 · 2025-11-05

**Soundness:** 3
**Presentation:** 3
**Contribution:** 3
**Rating:** 4
**Confidence:** 2

**Summary:**

The paper addresses critical flaws in current methods for evaluating Large Language Model (LLM) agents in cybersecurity tasks. It argues that existing benchmarks, which often reuse old Capture The Flag (CTF) challenges, are unreliable due to data contamination (where solutions are in the LLMs' training data) and potential cheating (where agents use web search to find public solutions).

To solve this, the authors introduce CTFUSION, a new streaming evaluation framework that uses LIVE CTFS—ongoing competitions with fresh, unreleased challenges. This approach prevents agents from finding existing solutions online. CTFUSION uses virtualization and aggregation to allow multiple agents to compete independently in live events without disrupting the real competition.

Experiments showed that agents performed significantly better on a static benchmark (14.4% success) than on LIVE CTFS (6.3%), suggesting the static results are inflated. A custom agent with web search achieved a 24.07% success rate on the static benchmark, confirming that cheating is a significant issue. The paper concludes that CTFUSION is a more robust solution for evaluating cybersecurity agents.

**Strengths:**

Demonstrating Vulnerabilities: It provides evidence that existing static CTF benchmarks are vulnerable to both data contamination and potential cheating.

Proposing CTFUSION: It proposes and implements CTFUSION, a novel, real-time streaming benchmark system that evaluates agents using LIVE CTFS to ensure challenges are new.

Providing Robust Evaluation: Through experiments, it shows that CTFUSION serves as a more robust and reliable solution for evaluating cybersecurity agents compared to static benchmarks.

Open-Sourcing: The authors are releasing the CTFUSION framework as open source to encourage future research in this area.

**Weaknesses:**

The paper's primary evidence is the performance gap between the static NYU CTF BENCH (14.4% success) and LIVE CTFS (6.3% success). The authors attribute this gap to two factors: (1) data contamination on the static benchmark and (2) task difficulty. The authors suspect contamination is the primary driver, but they provide no direct evidence to disentangle it from task difficulty. The conclusion that static benchmarks "inflate success rates" is an interpretation, not a proven fact. It is equally plausible that the five LIVE CTFs were simply harder or contained different types of problems than the 2017-2023 challenges aggregated in the static benchmark.

The paper introduces "D-CIPHER-WEB" to demonstrate "potential cheating." This experiment effectively proves that an agent with web access can find solutions that are publicly available. However, the paper creates a contradiction.

The paper evaluates agents on five LIVE CTFs. The resulting success rates are extremely low (averaging 5.11% to 7.10%). While the total number of problems across five CTFs (16+31+55+37+ScriptCTF) seems large, the number of successful solves—the key data point—is dangerously small. A 6.3% average success rate across ~170 problems means the entire conclusion about live performance is based on approximately 10-11 solved challenges per agent configuration. A difference of just one or two solved problems could dramatically swing the percentages (e.g., from 6.3% to 7.5%). The results are therefore highly sensitive to a small number of successes, making broad conclusions about the "true" capabilities of agents unreliable.

**Questions:**

See above

---

> ### Author Response · Authors · 2025-11-19
>
> We are grateful to the reviewer for the careful reading of our paper and for the balanced assessment that acknowledges both the strengths of CTFUSION and the limitations of our current evaluation. The comments on disentangling data contamination from task difficulty and on the small number of successful solves are particularly valuable, and we address these concerns in detail below.
>
>
> ### Response to Weaknesses1
> - We quantify difficulty using the standard CTFtime competition weight metric, which is computed from global participation and ranking statistics and is widely used in the CTF community as a proxy for event difficulty. For the NYU-CTF-Bench events, these weights range from 21.8 to 59, with a mean of 31.64. For the five LIVE CTFs studied in this work, the weights range from 24.48 to 69.35, with a mean of 39.64. According to this metric, LIVE CTF is on average somewhat more difficult, but the two distributions occupy a similar overall difficulty regime. Thus, while we do not claim that the difficulty is identical, a roughly 2× performance gap remains clearly large even after taking difficulty into account. We have incorporated these details into both the main text and the appendix.
> - At the same time, we fully acknowledge that difficulty and contamination cannot be perfectly disentangled. The revised paper makes this limitation explicit and clarifies that our claim is not that difficulty plays no role, but rather that contamination appears to be the primary and empirically supported factor.
>
> ### Response to Weaknesses2
> We agree that live success rates are computed from relatively few solves, making percentages sensitive to small changes. To address this:
>
> **Sensitivity checks.** We show how the live percentage changes under ±1–2 solved tasks and include leave-one-event-out re-computations. Even under these perturbations, the static–live gap remains large:
>
> - **Baseline:** 14.4% vs. 6.3% → ≈2.3× reduction
> - **Adding +2 solves to live (≈7.6%):** 14.4% vs. 7.6% → ≈1.9
> - **Removing –2 solves (≈5.3%):** 14.4% vs. 5.3% → ≈2.7
> The direction and order of magnitude (≈2× drop) are stable.

---

### Author Response · Authors · 2025-12-03

We thank the reviewers for their time and constructive feedback.
Below we briefly summarize our work and highlight what we see as its main strengths.

# Summary

The paper studies whether current CTF-based benchmarks provide trustworthy evaluation of LLM-based cybersecurity agents and introduces CTFUSION, a streaming framework that evaluates agents on unreleased challenges from LIVE CTF competitions while minimizing impact on real contests.

We empirically show that static CTF benchmarks can be both contaminated and cheatable. Augmenting D-CIPHER with web search (D-CIPHER-WEB) reveals concrete cheating behavior on NYU CTF BENCH, and a simple “no-cheat” ablation suggests that a substantial fraction of benchmark success comes from contamination-related behavior rather than genuine problem solving.

Using CTFUSION, we evaluate three LLMs, two agents, and five LIVE CTFs, finding that static NYU CTF BENCH scores are roughly twice higher than live performance while preserving the relative ranking of models and agents; we additionally reconstruct broken portions of NYU CTF BENCH and provide the full CTFUSION system and benchmark artifacts for community use.

# Key strengths

### Addresses an important and timely evaluation problem.
- Our work targets a concrete issue that affects many recent LLM cybersecurity papers: static CTF benchmarks can be contaminated by pretraining data and are vulnerable to on-the-fly cheating via web search or leaked artifacts. By documenting these risks in detail for a widely used benchmark, we provide direct evidence that current evaluation practice can substantially overestimate agent capability.

### Strong empirical evidence of cheating and contamination.
- We do not just hypothesize contamination; we instrument an existing agent to observe it in action. D-CIPHER-WEB nearly doubles performance on NYU CTF BENCH and exhibits clear behaviors such as copying flags from public write-ups, installing the nyuctf package, and directly reading flags from the dataset API. The “no-cheat” variant (D-CIPHER-NO-CHEAT) then shows a ~30% relative drop in success, isolating the impact of contamination-style behavior from genuine reasoning.

### Practical framework for LIVE CTF evaluation.
- CTFUSION is designed to be realistic and reusable. It uses a single CTFd account, per-agent virtualized views of challenge state, and a proxy that aggregates submissions and validates subsequent flags locally. This preserves contest integrity (no scoreboard distortion, low extra load) while allowing many agents to be evaluated independently. The implementation as an MCP server, plus simple environment-based configuration, makes it straightforward to plug in different agents (e.g., ENIGMA, D-CIPHER) and new CTFd-based events.

### Thorough and realistic experimental study.
- Rather than relying solely on a static dataset, we run agents in real 2025 online competitions (five LIVE CTFs across multiple organizers and challenge styles) and compare them to NYU CTF BENCH under the same protocols. We analyze not only overall success rates but also model vs. agent contributions, category-wise performance (e.g., pwn/crypto/forensics/misc/reversing), and detailed failure modes (cost limit, suspended, giveup). This provides a nuanced picture of where current agents actually fail in practice.

### High-impact artifacts for future evaluation.
- We reconstruct problematic NYU CTF BENCH tasks, provide a cleaned benchmark split, and release the full CTFUSION system (MCP server, proxy, control panel, and agent integrations).

We hope this summary helps the committee assess the contribution and potential impact of our work on trustworthy evaluation of LLM-based cybersecurity agents.

---

### Meta-Review · Area_Chair_aku5 · 2025-12-16

**Summary:**

This paper demonstrates that existing static CTF benchmarks for LLM-based cybersecurity (CTF) agent evaluation are vulnerable to (1) data contamination and (2) cheating via web search, and proposes CTFusion, a streaming evaluation framework using LIVE CTF. The engineering contributions and practical significance of the proposed evaluation platform are substantial, including the presentation of cheating instances via D-CIPHER-WEB log analysis, the quantification of performance degradation using D-CIPHER-NO-CHEAT, and the implementation of an MCP server on CTFd.

However, the decomposition of factors contributing to the performance gap between static benchmarks and LiveCTF (contamination vs. difficulty differences) is not fully resolved. Furthermore, due to the limited statistical stability of the LiveCTF side (insufficient repeated trials/variance estimation, low number of successes) and limited empirical backing for the claim that the "impact on actual contests is negligible," uncertainty remains regarding the generalization of the current conclusions.

**Reviewer Concerns:**

**Points Strengthened by Rebuttal/Author Comments**
* **Demonstration of Cheating Reality** (N8G9, xRZ7): Based on D-CIPHER-WEB logs, the authors showed specific behaviors such as retrieving write-ups/flags directly, increasing the persuasiveness regarding the structural risks of static/public benchmarks.
* **Partial Quantification of Contamination Contribution** (N8G9, bsbd, xRZ7): Results were added showing that pass@3 dropped with D-CIPHER-NO-CHEAT (prompt instruction change only), partially reinforcing that "contamination/dependence on known solutions cannot be ignored."
* **Clarification of Evaluation Settings** (xRZ7, NbCu): The authors clarified the sensitivity of pass@k and stopping conditions, explicitly stated that results should be interpreted as point estimates due to insufficient repeated trials, and outlined a policy for adding failure factor/category analysis.

**Remaining Concerns**
* **Incomplete Factor Disentanglement** (N8G9, bsbd, xRZ7): It remains insufficiently clear whether the gap between static vs. LiveCTF is primarily due to contamination or due to differences in difficulty/structure on the LiveCTF side (controlled conditions are limited, and additional experiments are noted as ongoing).
* **Statistical Stability** (N8G9, bsbd, xRZ7): Since LiveCTF has a limited number of problems per event and the number of solved problems is small, it is difficult to judge the strength of differences or generalization without estimated variance/confidence intervals.
* **Insufficient Empirical Measurement of Operational Impact** (xRZ7): While claims such as "negligible impact on competition" and "minimal setting changes" are intuitively reasonable, quantitative measurements of load/latency are not sufficiently presented.
* **Depth of Insights from LiveCTF** (bsbd, xRZ7, NbCu): While the addition of categorical/failure analysis is a step forward, it is currently limited, and a more systematic analysis/comparison is desired.
* **Generalizability (Model Selection)** (NbCu): The evaluation focuses on closed commercial models; control experiments using open-weight models remain a future task.
* **Limited Novelty** (xRZ7): The point remains that while valuable as an evaluation paradigm/infrastructure, technical novelty is limited.

**Reviewer Scores:**

* **Reviewer N8G9**: Main concerns are the "unseparated contamination vs. difficulty difference" and "small and unstable success counts in LiveCTF." Although the authors partially reinforced this with NO-CHEAT, the root concerns regarding factor decomposition and statistical stability remain. **Maintains 4 -> 4** (However, the weight is low as this reviewer's confidence=2).
* **Reviewer bsbd**: Doubts the representativeness of D-CIPHER-WEB (leaning towards "upper-bound cheating") and is skeptical about the academic depth and statistical treatment of LiveCTF. The authors clarified the positioning, but the fundamental stance is unlikely to change. **Maintains 2 -> 2** (High certainty).
* **Reviewer NbCu**: Highly values the engineering completeness and the importance of the problem setting, while requesting ranking correlations, model expansion, and failure analysis. The authors indicated stability in rankings and a policy for adding failure analysis, so the overall assessment did not shift significantly. **Maintains 6 -> 6** (Medium-High certainty).
* **Reviewer xRZ7**: Issues raised regarding the lack of actual measurement of "competition impact," factor decomposition, variance/retries, and novelty. The authors explained settings and indicated plans for partial additional analysis, but decisive factors (measurements and controlled experiments) are lacking. **Maintains 4 -> 4** (Medium certainty).

**Predicted Average:** (4 + 2 + 6 + 4) / 4 = **4.0**

---

### Decision · Program_Chairs · 2026-01-26

Reject